# KRAS-Dependency in Pancreatic Ductal Adenocarcinoma: Mechanisms of Escaping in Resistance to KRAS Inhibitors and Perspectives of Therapy

**DOI:** 10.3390/ijms24119313

**Published:** 2023-05-26

**Authors:** Enrico Gurreri, Giannicola Genovese, Luigi Perelli, Antonio Agostini, Geny Piro, Carmine Carbone, Giampaolo Tortora

**Affiliations:** 1Medical Oncology, Fondazione Policlinico Universitario Agostino Gemelli, IRCCS, 00168 Rome, Italy; enrico.gurreri01@icatt.it (E.G.); antonio.agostini@unicatt.it (A.A.); geny.piro@policlinicogemelli.it (G.P.); giampaolo.tortora@policlinicogemelli.it (G.T.); 2Department of Genitourinary Medical Oncology, The University of Texas MD Anderson Cancer Center, Houston, TX 77025, USA; ggenovese@mdanderson.org (G.G.); lperelli@mdanderson.org (L.P.); 3Department of Genomic Medicine, The University of Texas MD Anderson Cancer Center, Houston, TX 77025, USA; 4David H. Koch Center for Applied Research of Genitourinary Cancers, The University of Texas MD Anderson Cancer Center, Houston, TX 77025, USA; 5Translational Research to Advance Therapeutics and Innovation in Oncology, The University of Texas MD Anderson Cancer Center, Houston, TX 77025, USA; 6Medical Oncology, Università Cattolica del Sacro Cuore, 00168 Rome, Italy

**Keywords:** pancreatic cancer, KRAS, KRAS inhibitors, KRAS-dependency, resistance, escaping

## Abstract

Pancreatic ductal adenocarcinoma (PDAC) is still one of the deadliest cancers in oncology because of its increasing incidence and poor survival rate. More than 90% of PDAC patients are KRAS mutated (KRASmu), with KRASG12D and KRASG12V being the most common mutations. Despite this critical role, its characteristics have made direct targeting of the RAS protein extremely difficult. KRAS regulates development, cell growth, epigenetically dysregulated differentiation, and survival in PDAC through activation of key downstream pathways, such as MAPK-ERK and PI3K-AKT-mammalian target of rapamycin (mTOR) signaling, in a KRAS-dependent manner. KRASmu induces the occurrence of acinar-to-ductal metaplasia (ADM) and pancreatic intraepithelial neoplasia (PanIN) and leads to an immunosuppressive tumor microenvironment (TME). In this context, the oncogenic mutation of KRAS induces an epigenetic program that leads to the initiation of PDAC. Several studies have identified multiple direct and indirect inhibitors of KRAS signaling. Therefore, KRAS dependency is so essential in KRASmu PDAC that cancer cells have secured several compensatory escape mechanisms to counteract the efficacy of KRAS inhibitors, such as activation of MEK/ERK signaling or YAP1 upregulation. This review will provide insights into KRAS dependency in PDAC and analyze recent data on inhibitors of KRAS signaling, focusing on how cancer cells establish compensatory escape mechanisms.

## 1. Introduction

Pancreatic ductal adenocarcinoma (PDAC) is still one of the deadliest cancers in oncology and, due to its increasing incidence and overall five-year survival rate of less than 5% [1], it is expected to be the second leading cause of cancer-related death in the US by 2030 [2,3]. In addition, the annual incidence of PDAC is increasing in people younger than 30 years of age [4].

Among modifiable risk factors, current cigarette smoking, alcohol use, chronic pancreatitis, and obesity have strong associations with PDAC [3].

In most cases, PDAC is diagnosed at an advanced stage, locally advanced (30–35%) or metastatic (50–55%) [2], and treated with polychemotherapy regimens, including FOLFIRINOX, Gemcitabine/Nab-Paclitaxel, and nanoliposomal Irinotecan/Fluorouracil, with a survival benefit of 2–6 months compared with a single-agent Gemcitabine [3,5,6,7].

The tumor microenvironment (TME) plays a central role in PDAC biology and represents a desmoplastic scaffold characterized by intricate cellular and acellular crosstalk between activated cancer-associated fibroblasts (CAFs), tumor-associated macrophages (TAMs), myeloid-derived suppressive cells (MDSCs), regulatory T cells (Tregs), and bioactive specialized extracellular matrix (ECM), with low numbers of tumor-infiltrating lymphocytes (TILs). In this scenario, cell–cell interactions are obstructed by the ECM, which explains not only chemoresistance, but also poor response to immunotherapy.

In the heterogenous mutational landscape of pancreatic cancer, *KRAS*, *TP53*, *SMAD4*, and *CDKN2A* represent major oncogenic events involved in key molecular pathways such as DNA damage repair, cell cycle regulation, TGF-β signaling, chromatin regulation, and axonal guidance [8].

About 30% of all human cancers bear activating rat sarcoma (*RAS*) mutations; in particular, *Kirsten rat sarcoma* (*KRAS*) mutations are considerably more frequent than *Harvey rat sarcoma virus oncogene* (*HRAS*) and *Neuroblastoma RAS Viral Oncogene Homolog* (*NRAS*) mutations. As for PDAC, more than 90% of patients are *KRAS* mutated (KRASmu), and KRASG12D and KRASG12V are specifically the most common mutations [9].

The KRAS protein is a molecular switch that cycles between an active, Guanosine-5′-triphosphate (GTP)–bound state and an inactive, Guanosine-5′-diphosphate (GDP)–bound form. In cancer tumorigenesis, *KRAS* mutations typically increase the steady-state levels of the active form, driving protumorigenic pathways, such as the mitogen-activated protein kinase (MAPK) and Phosphatidylinositol 3-kinase (PI3K) pathways.

The first genetic event that leads the earliest precancerous lesions to invasive pancreatic cancer is the mutational activation of KRAS [10,11]. Despite this critical role, its high affinity for nucleotide and the lack of viable binding pockets for small-molecule inhibitors have made direct targeting of the RAS protein extremely difficult over the past four decades [9,12]. In this context, understanding PDAC tumorigenesis is crucial for both the identification of early diagnostic markers and the development of multiple alternative modes of intervention.

In this review, we explore KRAS-dependency in PDAC and analyze recent data on KRAS signaling inhibitors, focusing on how cancer cells establish compensatory escape mechanisms.

## 2. KRAS-Dependent Tumorigenesis in PDAC

Several studies have demonstrated the strong association between PDAC and inflammation. In the context of chronic pancreatitis, the inflammatory microenvironment can activate survival and proliferation programs and induce chromatin changes in cancer cells, promoting tumor growth. Oncogenic *KRAS* accelerates this process in pancreatic tissue (Figure 1) [13,14,15], inducing, along with inflammatory damage (e.g., cerulean-induced pancreatitis) and other tumor suppressor deficiencies (e.g., protein (p)16INK4a/p14ARF, Tumor Protein p53 (TP53) and/or Suppressor of Mothers against Decapentaplegic 4 (SMAD4)) loss, the appearance of neoplastic precursor lesions, such as acinar-to-ductal metaplasia (ADM) and pancreatic intraepithelial neoplasia (PanIN) [15].

### 2.1. KRASmu, Inflammation and Precursor Lesions

KRASmu expression is not sufficient to initiate tumorigenesis in the pancreas, and for ADM induction and development of pancreatic acinar cells, further subsequent events are necessary, such as additional genetic lesions, chronic inflammation, or upregulation of growth factor signaling [11,16,17].

PanIN is considered the major pathological basis of PDAC development with properties of ductal cells as well as tumor cells in PDAC. However, recent data in engineered mouse models (GEMM) of pancreatic cancer have evidenced that acinar cells are the main cellular origin of PDAC. Acinar cells, through ADM, show increased expression of ductal cell markers such as cytokeratin-19 (CK-19) or sex-determining region Y box 9 (SOX9) and reduced expression of acinar cell markers, such as amylase or MIST-1 [18,19,20]. In the presence of KRASmu, the ADM process become irreversible and leads to a change in cellular identity (transdifferentiation) and progression to PanIN. Transcription factors controlling pancreatic duct development, such as SOX9 and hepatocyte nuclear factor 6 (HNF6), or other ones critical for somatic stem cell reprogramming, such as Kruppel-like factor 4 (KLF4), have been demonstrated to regulate ADM process [18]. Ge W and colleagues demonstrated that SOX9 and Phos-SOX9 (S181) levels in acinar cells are both regulated by miR-802, a pancreatic microRNA (miRNA), which controls ADM formation in the presence of oncogenic KRAS [21]. Costamagna A and colleagues demonstrated that integrin and growth factor receptor signaling converge on p130Cas, an adaptor protein encoded by *BRCA1* and a downstream effector of the KRAS pathway, to induce tumorigenesis and boost acinar to ductal metaplasia and subsequent tumorigenesis through PI3K activation [11].

The nuclear factor kappa-light-chain-enhancer of activated B cells (NFκB) family of transcription factors is composed of NFκB2 (p100/p52), NFκB1 (p105/p50), RelA/p65, RelB, and Rel. NFκB2 is necessary for KRASG12D-dependent ADM development, PanIN progression, and tumor proliferation. In the same way, RelB promotes PanIN progression in the KRASG12D PDAC cells [22].

Recent data have shown that *ring finger protein 43* (*RNF43*) is the most frequently mutated gene in IPMNs together with *KRAS*. RNF43 is a member of the RING finger protein family, and an E3 ubiquitin ligase. It mediates the ubiquitination and degradation of the Wnt receptor complex component frizzled, and chains transcription factor 4 (TCF4) to the nuclear membrane by silencing its transcriptional activity. Thus, the loss of RNF43 activates the Wnt/β-catenin signaling pathway [23].

RNA sequencing analysis of PDAC patients in recent works showed high expression of angiopoietin-like 4 (ANGPTL4), which is involved in epidermal differentiation and development of PDAC. Yan HH et al. revealed a tumor-promoting role of ANGPTL4 through regulation of periostin/integrin signaling during PDAC initiation and maintenance. These data suggested the ANGPTL4/periostin axis as a potential molecular target for the prevention of PDAC [24].

IL-33 has a central role in this program, bridging tissue damage with KRAS-dependent epithelial plasticity and accelerating the appearance of early precursor lesions (PanIN) after injury. Additionally, in early neoplasia this cytokine induces an immunosuppressive TME [25].

### 2.2. KRASmu and Metabolism

KRASmu PDACs depend on glucose and glutamine for energy production and maintenance of the redox balance and show decreased levels of intracellular amino acid. On the other hand, autophagy is required for the maintenance of KRASmu cells [26]. Moreover, recent studies showed how autophagy increases MHC degradation as an escape mechanism from T lymphocytes surveillance and makes tumor cells resistant to immunomodulatory drugs [27].

KRASmu pancreatic cancer cells can channel their glucose metabolism away from the mitochondria through programmed mitophagy via the mediator BCL2/adenovirus E1B 19-kDa-interacting protein 3-like (BNIP3L/NIX). NIX ablation in vivo has been shown to delay the progression of PanIN to PDAC. In this context, Viale A et al. demonstrated that quiescent KRASmu pancreatic cancer cells that survive oncogene ablation, which are responsible for tumor recurrence, have cancer stem cells characteristics and depend on oxidative phosphorylation (OXPHOS) for survival [28]. Glutamine metabolism is involved in redox homeostasis and plays a key role in tumor growth [29].

Previous work has highlighted the crucial role of branched-chain amino acid (BCAA) metabolism in metabolic adaptation in cancer. In particular, BCAA transaminase 1 (BCAT1) or BCAT2 has been shown to be upregulated and important for proliferation in PDAC [30].

### 2.3. KRASmu and TME

Mutant KRASG12D upregulates peroxisome proliferator activated receptor-delta (PPARδ) in human and murine PanIN lesions, promoting inflammation-related signaling pathways and pancreatic tumorigenesis. PPARδ hyperactivation recruits TAMs and MDSCs via the CCL2/CCR2 axis, remodeling the immune TME. This process has been demonstrated in mouse models fed with high-fat diets enriched with fatty acids that are natural ligands of PPARδ [31].

KRASmu upregulates IL2Rγ and IL4R, two members of the type I cytokine receptor family, the former of which contributes to PDAC tumorigenesis. In fact, the cytokines IL4 and IL13 signaling via IL4R drive JAK-STAT-cMYC activation, resulting in increased glycolysis and tumor growth [30]. Previous works have shown that cMYC expression enhances tumor cell proliferation [32]. In KRASmu PDACs are found higher levels of cytokines IL4, IL13, and, as mentioned in this work, IL-33, secreted by GATA-3+ TH2 polarized CD4+ T cells, resulting in immunosuppressive TME and a self-perpetuating cascade of pro-tumorigenic effects [31]. In addition, KRASmu is crucial for the crosstalk between PDAC cells and activated cancer-associated fibroblasts (CAFs). CAFs are activated through transforming growth factor beta (TGF-β) and sonic hedgehog pathways and regulate the tumor stroma, including extracellular matrix, collagen fibers, and hyaluronic acid, promoting PDAC cell growth and an immunosuppressive TME [10].

Moreover, oncogenic KRASG12D suppresses programmed death ligand 1 (PD-L1) expression and has the strongest suppressive power compared with other KRAS mutations [33].

### 2.4. KRASmu and Chromatin

In the injury-prone pancreas, oncogenic KRAS mutation induces an epigenetic program alternative to physiological regeneration, which leads to PDAC initiation. Alonso-Curbelo D et al. showed in vivo that pancreatic metaplasia is characterized by epigenetic silencing of acinar identity loci that is enhanced by bromodomain containing 4 (BRD4) suppression [25]. The bromodomain and extraterminal (BET) family member BRD4 is a chromatin reader which binds acetylated and active chromatin enhancing transcription of cell-identity genes.

In KRASmu cells, within 48 h of tissue damage, progression to neoplasia is facilitated by interactions between genetic and environmental insults leading to an ‘acinar-to-neoplasia’ chromatin switch that alters DNA accessibility. This chromatin remodeling program is present from the early stage of disease to metastasis in advanced PDAC [25].

Sangrador I et al. demonstrated that the transcriptional repressor zinc finger E-box binding homebox 1 (ZEB1) is a key mediator of KRAS-dependent oncogenesis in vivo; indeed, in the presence of a KRAS mutation, ZEB1 haploinsufficiency delays PDAC development. Notably, ZEB1 is predominantly expressed in stromal myofibroblasts associated with PanIN and PDAC [34]. Genovese G et al. highlight the crucial tumor-suppressive role of SWI/SNF related, matrix associated, actin dependent regulator of chromatin, subfamily B, member 1 (SMARCB1) as a differentiation checkpoint and a gatekeeper of epithelial–mesenchymal transition. This is a novel mechanism of KRAS-dependent tumorigenesis of PDAC cells that fails to activate downstream KRAS signaling (e.g., through MAPK) [35]. SMARCB1 is a switch/sucrose non-fermentable (SWI/SNF) chromatin remodeling factor whose activity restrains growth and metabolic programs via MYC activation [35].

## 3. KRAS Signaling Inhibitors in PDAC

### 3.1. KRASG12C Inhibition

The frequency of KRASG12C mutations in PDAC patients is abnormally high in some regions such as Japan, while its frequency in PDAC patients worldwide is quite low. However, none of the KRASG12C inhibitors, such as sotorasib or adagrasib, have been approved as a treatment for PDAC [36]. A previous work has demonstrated the growth inhibition power of Adagrasib (MRTX849) in a pancreatic cancer cell line [37,38] and it is undergoing clinical trials for patients with KRASG12C mutant pancreatic cancer (NCT03785249) (Table 1). Additionally, a confirmed partial response has been reported in the phase I/Ib cohort in a patient with PDAC (NCT03785249). Ostrem et al. pinpointed the druggable switch-II pocket in KRASG12C through X-ray crystallography and mass spectrometry [38]. Only a small percentage of PDAC patients harbor the G12C mutation; on the other hand, as previously mentioned in this review, G12D is the most prevalent KRAS mutation in PDAC.

### 3.2. KRASG12D Inhibition

Various groups have shown that it is possible to pharmacologize GTP-bound KRASG12D: KD2 is a cyclic peptide that can selectively target the switch-II groove in mutant GTP-bound KRASG12D; both in vitro and in vivo, KS-58, a bicyclic peptide, has shown activity against KRASG12D mutated pancreatic cancer [39]. MRTX1133 is a small molecule that selectively targets KRASG12D by blocking downstream pathways through inhibition of nucleotide exchange and binding of effector rapidly accelerated fibrosarcoma 1 (RAF1); in vivo, MRTX1133 reduced phosphorylation of extracellular signal-regulated kinase (ERK), resulting in tumor regression [33]. Using a medicinal chemistry approach, other compounds were discovered: TH-Z827 and TH-Z835 are two inhibitors that bind with Asp12 inside the switch-II pocket, specifically inhibiting KRAS signaling, and not KRASG12C or WT, in G12D mutant PDAC in vitro and in vivo; in vitro, KD-8 is another inhibitor of KRASG12D PDAC tumor growth [33]. These works provide proof of concept evidence that the KRASG12D mutation could potentially be targeted, benefiting a larger number of PDAC patients [38].

### 3.3. Other Inhibitors

RNA interference (RNAi) has been demonstrated to suppress KRASmu expression in pancreatic cancer cells inhibiting anchorage-independent growth and tumorigenic proliferation. These effects suggest RNAi as a potential drug for KRASmu PDAC [10]. Messenger ribonucleic acid (mRNA)-5671/V941 is a novel mRNA vaccine encoding mutant KRAS which is being studied in patients with KRASmu cancers in phase I clinical trial (NCT03948763), with or without Pembrolizumab. Similarly, dendritic cell vaccines (NCT03592888) and peptide vaccines (NCT04117087) for KRASmu patients are in clinical trials (Table 1) [38].

Engineering patient lymphocytes to express receptors that specifically target tumor neoantigens is known as adoptive cell therapy [40]. A previous work has generated murine T cells that can recognize KRASG12D PDAC in an HLA-A*11:01 restricted manner and inhibit tumor growth in vivo [41]. Two ongoing phase I/II clinical trials are investigating the transfer of such cells engineered to express the murine T-cell receptor (TCR) specific for KRASG12D (NCT03745326) or KRASG12V (NCT03190941) (Table 1) in an HLA-restricted manner, in patients with solid tumors, including KRASG12D PDAC [33,38].

Tricomplex inhibitors bind cyclophilin A, a chaperone protein ubiquitously present inside the cell [42], which in turn binds the target protein, creating a target-inhibitor-cyclophilin-A complex. RMC-9805 is a novel specific tricomplex inhibitor of KRASG12D which suppresses tumor growth in xenograft PDAC [33]. RMC-6236, on the other hand, is a multi-RAS inhibitor tricomplex, capable of targeting multiple different mutations of KRAS, such as KRASG12V [38].

For KRAS to localize on the membrane and become active, giving rise to signal transduction, its post-translational prenylation catalyzed by farnesyl transferases (FTases) is required [43]. For this reason, a number of FTase inhibitors (FTIs), e.g., lonafarnib and tipifarnib, have been designed and clinically studied [44], reaching phase III clinical trials for various cancer types, with disappointing results in PDAC [38].

## 4. Mechanisms of Escaping in Resistance to KRAS Inhibitors

Despite promising results in preclinical and clinical studies of several inhibitors outlined in this paper, there are still few studies on mechanisms of escaping in resistance to KRASG12D inhibitors. So far, even MRTX1133, a KRASG12D inhibitor, has not entered clinical trials yet. Thus, most of the data about mechanisms of escaping in resistance to KRASG12D need to be translated from studies upon clinical application of KRASG12C inhibitors.

Pre- and post-treatment comparison of samples from patients treated with the KRASG12C inhibitor sotorasib showed multiple genetic escape mutations after treatment in 63% of patients, including *KRAS*, *NRAS*, *BRAF*, *EGFR*, *FGFR2*, and *MYC* [36]. Further exploration revealed that the inhibitory effect of sotorasib was reduced after the occurrence of KRASG12V, NRASQ61K or MRASQ71R (a small GTPase regulating the dimerization and activation of CRAF, a RAF family protein in KRAS downstream of the MAPK-ERK pathway [36]). The binding site for a series of KRASG12C inhibitors, including sotorasib, is the cysteine 12 residue of the KRASG12C protein, which is located near the switch-II pocket [45]. Mutations at this site, such as KRASY96D found in a patient after treatment with the KRASG12C inhibitor MRTX849, disrupt the hydrogen bond between the binding site and KRASG12C inhibitor [36]. In the development of resistance, other acquired genetic mutations in *KRAS* are G12D/R/W, G13D, Q61H, R68S, H95D/Q/R, Y96C, and KRASG12C allele amplification, as well as genetic changes such as *MET* (also known as the N-methyl-N0-nitroso-guanidine human osteosarcoma transforming gene) amplification, mutational activation of mitogen-activated protein kinase kinase 1 (MAP2K1) also known as *MEK* and rearranged during transfection (*RET*), oncogene fusion of anaplastic lymphoma kinase (*ALK*), *RET*, *B-RAF*, *RAF1*, and fibroblast growth factor receptor 3 (*FGFR3*), loss-of-function mutations in neurofibromatosis 1 (*NF1*) and phosphatase and tensin homolog (*PTEN*) [12,38]. These mutations increase the levels of active GTP-bound KRAS protein, preventing drug binding [46]. It has been demonstrated that KRASmu inhibitors, such as sotorasib, or MEK inhibitors, such as trametinib, to induce escape mechanisms in PDAC, cause activation of the mechanistic target of rapamycin 2 (mTORC2) molecule rapamycin-insensitive companion of mammalian target of rapamycin (RICTOR) and phosphorylation of AKT at Ser-473 by integrin-linked kinase (ILK) in several PDAC mouse models and human tumors [38]. After these results, it was shown that inhibition of mTORC2 alone stimulates ERK activation [47], promoting cell survival. On the other hand, mTORC2 signaling has an important role in the development of resistance in PDAC; in fact, its inhibitors have anti-tumor activity in PDAC cells when combined with an inhibitor of KRASmu or MEK [38]. Because PDAC is characterized by high cell heterogeneity, it is highly likely that future use of KRASG12D inhibitors in PDAC may lead to the same result by increasing mutational burden with the previously mentioned mutations and genetic changes [36].

Cheng DK et al. demonstrated that oncogenic KRASmu inhibits wild type (WT) RAS signaling through NF1/Ribosomal S6 kinase 1 (RSK1) [48]. In fact, the inhibition of this negative feedback pathway with KRAS inhibitors activates WT RAS signaling and promotes adaptive resistance, as evidenced by PDAC cells that survived after treatment with KRASG12V inhibitor. In addition, inhibition of both WT RAS, through son of sevenless 1 (SOS1) inhibition, and KRASG12C mutation, through AMG 510, showed the best response in vitro, demonstrating a synergy between KRASG12C inhibitors and upstream effectors (SOS1) inhibitors. These data strengthen the idea of WT RAS as a central actor in the acquired resistance to KRASmu inhibitors [49].

Alterations of multiple receptor tyrosine kinase (RTK)-RAS-MAPK pathways contribute to the resistance to KRASG12C inhibitor adagrasib, and the combination of inhibitors of RTK, Src homology region 2-containing protein tyrosine phosphatase 2 (SHP2), and KRASmu is the subject of ongoing clinical trials (NCT04330664, NCT04185883) (Table 1) [12,38].

Moreover, KRASmu tumors have another intrinsic mechanism of resistance; in fact, in the presence of FTIs described above, KRAS is prenylated by geranylgeranyl transferase 1 (GGTase1), the so-called alternative prenylation [38].

As previously stated in this paper, the TME is composed of various types of cells and plays a central role in chemoresistance and immunoresistance in PDAC. CAFs have been shown to contribute to resistance to therapies such as KRASmu inhibitors, and to express vitamin D receptor, which plays an important role in the development of resistance; confirming this, a work has shown how calcipotriol, a vitamin D receptor agonist, can override the influence of CAFs in a murine model [38]. As stated above, autophagy is necessary for the maintenance of KRASmu cells; in fact, cancer cells utilize several metabolites derived from autophagic degradation of CAFs, resulting in resistance to different therapies including KRAS inhibitors [50]. As well as CAFs, neurons present in the TME release amino acid serine, promoting PDAC growth [38].

## 5. Alternative Targets in KRAS Signaling Pathways and Future Perspectives

### 5.1. Alternative Targets in KRAS Signaling Pathway

An effective strategy to overcome resistance to KRAS inhibitors is combining different types of therapies, e.g., the combination of KRASG12C inhibitors with SHP2 (an upstream effector of RAS) inhibitors has been shown to surpass resistance to KRASG12C inhibitors alone and remodel TME in PDAC models, reducing the number of activated CAFs [49]. On the other hand, the combination of KRAS inhibitors with downstream target therapies, such as MAPK-ERK and PI3K-AKT-mammalian target of rapamycin (mTOR) pathways inhibitors, showed disappointing results in pancreatic cancer models. Nevertheless, the inhibitory effect of histone deacetylase (HDAC) synergizes with the combined targeting of the MAPK-ERK and PI3K-AKT-mTOR pathways [36]. Promising new combinations include KRASG12C inhibitors with cell cycle checkpoints or immune checkpoints inhibitors (ICIs) [37,45], or the triple inhibition of KRASG12C/SHP2/PD-L1, tested in PDAC murine models [51]. Recent data suggest nuclear export protein exportin 1 (XPO1), which transports protein cargo from the nucleus to the cytoplasm, as an important factor in relieving tumor cells from resistance to KRASG12C inhibitors [36]. 

Another promising factor in resistance to KRAS inhibition is the deubiquitinase ubiquitin specific peptidase 21 (USP21), which at is amplified and overexpressed in about 20% of PDAC patient samples. Its nuclear activity promotes pancreatic tumor growth and tumor stem cell properties. USP21 deubiquitinates the transcription factor 7 (TCF7) and amplifies canonical Wnt signaling [52]. Further explorations are needed to test USP21 inhibitors with or without KRAS inhibitors as possible therapies in PDAC [53].

Different molecules that prevent the interaction between son of sevenless (SOS1) and KRAS have been identified [38]. SOS1 is a guanine nucleotide exchange factor (GEF) that binds GDP-bound KRAS and catalyzes the switch of GDP for GTP, activating KRAS [54]. Its low levels result in inhibition of tumor growth. BI-3406 is an SOS1 inhibitor active only in KRASmu cells with anti-tumoral activity synergistic with MEK inhibitors. BI-1701963 is another SOS1 inhibitor which is being studied in an ongoing phase I clinical trial (NCT04111458) with or without MEK inhibitor (trametinib) in patients with KRASmu cancers (Table 2) [38].

SHP2 is a tyrosine phosphatase protein encoded by the gene *PTPN11* with an intrinsic regulatory mechanism [38]. Its role in KRAS signaling seems to be linked to other proteins such as SOS1 and growth factor receptor-bound protein 2 (GRB2) and it is involved in various pathways such as KRAS-MAPK signaling [55]. SHP2 plays a central role in cancer development in KRASmu PDAC and NSCLC models. Its inhibition has been seen as synergistic with MEK inhibition, stopping tumor growth in PDAC and NSCLC models in vivo [38]. SHP099 is a compound that locks SHP2 in its inactive, autoinhibited state and is able to inhibit tumor growth via the MAPK pathway in vivo [56]. Several potential inhibitors of SHP2 are in different ongoing clinical trials [57]: RMC-4630 is currently in phase I/Ib clinical trials with or without an ERK inhibitor (NCT03634982, NCT04916236); RMC-4550 has shown to inhibit KRASmu cells proliferation in preclinical models [58]; TNO155, an allosteric inhibitor of SHP2 with demonstrated anti-tumoral activity through MAPK, is in several ongoing phase I and II clinical trials with or without several synergistic targeted therapies (NCT04000529, NCT04330664, NCT03114319) (Table 1 and Table 2) [59]; other SHP2 inhibitors are in clinical trials, such as ERAS-601 (NCT04670679), JAB-3312 (NCT04121286), BBP-398 (NCT04528836), and RLY-1971 (NCT04252339) (Table 2) [38].

### 5.2. Future Perspectives

Proteolysis targeting chimeras (PROTACs) consist of two peptides, one that binds the target protein linked to another peptide that recruits an E3 ubiquitin ligase for proteasomal degradation of the target protein [60]. LC-2 is a PROTAC that targets KRASG12C composed of MRTX849 bound to a von Hippel–Lindau (VHL) recruiter peptide; the former binds covalently to KRASmu, and the latter induces sustained proteasomal degradation of KRASmu and subsequent MAPK inhibition. The compound 17f is a PROTAC that targets phosphodiesterase 6 (PDEδ), an important prenylation factor [38].

NS-1 is a monobody inhibitor of RAS dimerization that targets the α4–α5 interface of KRAS, able to prevent the development and progression of pancreatic cancer in murine model. The unique problem with this potential new strategy is the size of the molecule NS-1, which makes its intracellular localization difficult [38].

Peptide nucleic acids (PNAs) are synthetic nucleotide analogs capable of binding to specific complementary DNA and RNA sequences [36] or to the mRNA of the target gene, inhibiting its translation process [61]. In a previous work, PNAs significantly inhibited tumor cell activity and reduced KRASG12D gene expression in the human metastatic pancreatic adenocarcinoma cell line AsPC-1 [36].

As previously mentioned in this paper, WT KRAS can induce resistance to KRAS inhibitors with its compensatory effect, and in this setting pan-RAS inhibitors could overcome this mechanism. Jin Wang et al. have developed several small-molecule pan-RAS inhibitors that stabilize the “open non-signaling intermediate conformation” of RAS [62]: NSC290956 (also called Spiclomazine or APY606) inhibited the proliferation of KRAS-dependent pancreatic cancer cell lines CFPAC-1 (KRASG12V), MIA PaCa-2 (KRASG12C), Capan-1 (KRASG12V), SW1990 (KRASG12T), and BxPC-3 (WT KRAS) [36]; NSC48160 showed similar effect on CPFAC-1 (KRASG12V) and BxPC-3 (WT KRAS) [63] and induced apoptosis in MIA PaCa-2 (KRASG12C) [64]; inhibitory effects of NSC48693 on KRAS-dependent cancer cells were superior to those of NSC48160 on CFPAC-1(KRASG12V), MIA PaCa-2 (KRASG12C), and BxPC-3 (WT KRAS) cells [36].

We have already mentioned a class of tricomplex inhibitors that has shown promising results; another similar class of compounds is the tricomplex RAS-ON (RASON) inhibitors, comprising KRASG12C, G12D, and G13C inhibitors, and a G12X inhibitor which targets multiple different G12 mutations, which are currently being studied after initial preclinical data [65].

As mentioned earlier in this review, IL4/IL13 cytokines play a central role in TME remodeling via the IL4-IL4R-Janus kinase (JAK)-signal transducer and activator of transcription proteins (STAT) signaling cascade. However, trials with JAK1/2 inhibitors have yielded disheartening results [66]. Further explorations are necessary to study interactions between cancer cells and TME cells.

We have already discussed RNAi, which still poses a challenge because of enzymatic breakdown, renal clearance, and precise targeting of the tissue of interest. Loaded siRNAs targeting KRAS into local drug eluter (LODER), a biodegradable polymetric matrix that protects siRNA allowing constant local release of siRNA inside tumor tissues, have shown anti-tumoral activity towards human pancreatic tumor cells by increasing the survival of murine models. In an open-label phase I–IIa study with 15 patients enrolled, LODER siRNAs in combination with FOLFIRINOX showed a median overall survival rate of 15 months and an 18-month survival rate of 38.5%. A phase II study of patients with locally advanced PDAC is ongoing to test siRNAG12D LODER in combination with standard therapy [10]. Another approach for siRNA delivery against KRASmu is represented by exosomes, also termed inhibitory exosomes (iExosomes), which have shown efficacy in several preclinical models of pancreatic cancer [10]. In addition, CD47, a ‘do not eat me’ signal present on iExosomes, enables them to enter cells via micropinocytosis that is enhanced by KRASmu-dependent tumorigenesis. A clinical trial is pending to address the feasibility, safety, and efficacy of iExosomes in patients with metastatic pancreatic cancer [10,67]. One, 2-dioleoyl-sn-glycero-3-phosphatidylcholine (DOPC) is the major component of a promising nanoliposomal platform used in vivo for KRAS-targeting siRNA delivery [10].

Recent data have already challenged conventional knowledge that *KRAS* mutations (e.g., G12C, G12D, G12V or G12S) cannot hydrolyze GTP and return to the GDP-bound state and have demonstrated that these mutations are still able to hydrolyze GTP, suggesting a further fine-tuning regulation of RAS. The aforementioned RASON, a novel protein encoded by the long non-protein intergenic coding RNA 00673 (LINC00673), is the first identified positive regulator of KRAS that binds directly to it, stabilizing its hyperactive state in a way that differs from guanine nucleotide exchange factors (GEFs) or GTPase-activating proteins (GAPs) [9].

As we stated above, the BET family member BRD4 is a chromatin reader which binds acetylated and active chromatin, enhancing transcription of cell-identity genes with a tumor-permissive role. Principe et al. have developed XP-524, a promising BET inhibitor that has shown encouraging results in combination with gemcitabine or PARP inhibitors by restraining the effects of *KRAS* activating mutations. In addition, XP-524 increases CD4+ and CD8+ T cell populations [68], suggesting a possible strategy to sensitize KRASmu PDAC to immune checkpoint inhibition.

Jonghwa Jin et al. explored glutamine metabolism in PDAC, targeting glutamine transporters as a promising strategy for advanced or drug-resistant cancers [29].

Dixon defined ferroptosis as a type of iron-dependent non-apoptotic cell death in KRASmu cancer cells in 2012 [69], however oncogenic KRAS makes cells more resistant to ferroptosis through upregulation of ferroptosis suppressor protein 1 (FSP1). Based on these data, Müller F and colleagues suggested ferroptosis induction combined with FSP1 inhibition as a new therapeutic strategy against KRASmu cancers [70].

## 6. KRAS Dependency in PDAC

KRAS regulates development, cell growth, epigenetically dysregulated differentiation, and survival in PDAC through activation of key downstream pathways, such as MAPK-ERK and PI3K-AKT-mTOR signaling, in a KRAS-dependent manner [71]. Previous works showed that some pancreatic cancer cell lines survive to KRASmu silencing and then depend on PI3K. This mechanism induces overexpression of yes-associated protein 1 (YAP1), an important transcriptional co-activator of the hippo pathway, thus escaping KRAS inhibition [36]. Furthermore, higher dosage of the mutant allele of *KRAS* than its WT counterpart has been associated with poor prognosis in cancer patients. This phenomenon is called “mutant allele-specific imbalance” (MASI) [71]. In addition, recent data have shown that restoration of WT KRAS in pancreatic cancer cells induces inhibition of nuclear translocation of YAP1 [71,72], which is associated with poor prognosis in PDAC patients [73]. In mouse models, after KRAS inactivation, one-third of spontaneous tumor recurrences that escape KRASG12D dependency shows deficit of KRAS expression and YAP1 amplification, resulting in an aggressive quasi-mesenchymal phenotype with the activation of cell cycle and DNA repair pathways through cooperation of the transcription factor E2F [10].

Chen et al. observed that murine PDAC cells, when KRAS is sustainedly silenced, undergo a reversible cell state change without mutational or transcriptional alterations, characterized by morphological changes and tumor-promoting activity with activation of the focal adhesion pathway, suggesting that the latter is a possible manifestation of acquired KRAS independency [38].

USP21’s ability to bypass KRAS dependency occurs in the cytoplasm with a novel mechanism different from the previously mentioned KRAS escape mechanisms, such as activation of MEK/ERK signaling or YAP1 upregulation. Hou et al. evidenced that USP21 reduces autophagy and increases amino acid levels, resulting in upregulation of MTOR-associated signaling [74]. Further analysis demonstrated that microtubule affinity regulating kinase 3 (MARK3), a microtubule-binding kinase and regulator of microtubule dynamics, is directly deubiquitinated by USP21, leading to KRAS-independent growth, cancer development, and macropinocytosis, a key metabolism mechanism for KRASmu PDAC cell survival.

Similarly, Hou et al. showed that an upregulation of HDAC5 enhances the recruitment of TAMs into the TME, promoting resistance to KRAS inhibitors through the activation of the C-C motif chemokine ligand 2 (CCL2)/C-C motif chemokine receptor 2 (CCR2) axis and of transforming growth factor-β (TGF-β) in a SMAD-4 dependent manner, bypassing KRAS-dependency [38,75].

Several clinical studies have shown that KRAS inhibition blocks both PI3K-AKT-MTOR and MAPK signaling in KRAS-dependent tumors [76]. On the other hand, inhibition of the MAPK pathway alone leads to hyperactivation of the PI3K-AKT-mTOR pathway via different RTKs such as AXL and Platelet derived growth factor receptor alpha (PDGFRa), and activation of several escape circuits, such as recruitment of insulin receptor kinase by MTORC1 and MEK inhibition-mediated hyperactivation of the ERBB receptors epithelial growth factor receptor (EGFR), human epidermal growth factor receptor 2 (HER2), and ERBB3. However, RTK activation after MEK inhibition has been demonstrated in both RASmu and WT RAS tumor models and recent data have evidenced that combinations of inhibitors are highly toxic [76]. 

Thus, KRAS dependency is so strong and essential in KRASmu PDAC that cancer cells have secured several compensatory escape mechanisms to counteract the effectiveness of KRAS inhibitors [77].

## 7. Conclusions

Pancreatic ductal adenocarcinoma remains a real challenge in oncology. A lot of new possibilities are now actual therapies, but more studies are needed to refine these new strategies.

In the past 30 years, around 70 reviews on KRAS’s role in PDAC have been published. We conducted this review because we felt it was necessary to collect the most up-to-date data covering the past 5 years and create a new comprehensive information base to rely on for future research. We sought to summarize and make the topic clear. We have focused on KRAS-dependency, enlightening this key role in PDAC tumorigenesis. We found that KRASmu influences cells at early stages of tumorigenesis in different ways. There are several oncogenic mutations of *KRAS*, and we focused on the more frequent ones in PDAC.

We reviewed all types of direct and undirect inhibitors of KRAS signaling, its upstream and downstream effectors, critically analyzing what is now affirmed as reality and current therapies. We showed how these new strategies are limited by several mechanisms of escape, highlighting the necessity of much more studies to understand how to overcome these limitations.

ADM and to a greater extent PanIN represent the critical elements in the establishment of oncogenic *KRAS* dependency and its further effects. In this setting, PPARδ is a potential target to prevent PanIN cancerization [31].

Chromatin remodeling plays a central role in identity changes of ADM in the inflamed and injured pancreas, suggesting a tool for early detection of epigenetically dysregulated programs in PDAC development [25]. These changes often lead to cancer stem cells characteristics and oxidative phosphorylation (OXPHOS) for cancer survival, showing other new viable targets [28]. Often, the PDAC phenotype changes to aggressive mesenchymal type, showing how SWI/SNF-controlled proteostasis, with its chromatin remodeler SMARCB1, need to be further explored to better understand the epithelial–mesenchymal transition in PDAC [35].

Several emerging therapies use inhibitors of different players in KRAS signaling, such as p130Cas [11], dual inhibition of FTase and GGTase activity [38], or RASON [9] in KRASmu PDAC.

However, the endless combination possibilities of inhibitors lead to infinite possibilities of mechanisms of escaping and resistance that we have yet to fully understand and overcome in order to definitively win against PDAC.

## Figures and Tables

**Figure 1 ijms-24-09313-f001:**
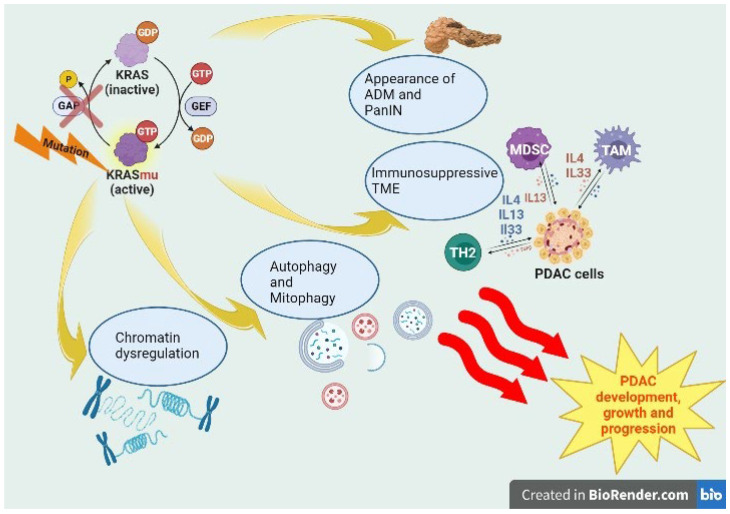
KRAS-dependent tumorigenesis in PDAC. In PDAC tumorigenesis, KRAS mutations typically increase the steady-state levels of the active form driving protumorigenic pathways. KRASmu negatively influences the regeneration program by inducing the appearance of neoplastic precursor lesions, such as acinar-to-ductal metaplasia (ADM) and pancreatic intraepithelial neoplasia (PanIN). Its influence on metabolism leads to fluxes of autophagy and mitophagy. Higher levels of cytokines IL4, IL13 and IL-33, secreted by GATA-3+ TH2-polarized CD4+ T cells, are found in KRASmu PDACs, resulting in an immunosuppressive tumor microenvironment (TME). Oncogenic KRAS mutation induces an epigenetic program, an alternative to physiological regeneration, which leads to PDAC initiation.

**Table 1 ijms-24-09313-t001:** Registered trials of KRASmu signaling inhibitor combination therapy on clinicaltrials.gov, accessed on 6 Marzh 2023.

ClinicalTrials.Gov Identifier	Title	Phase	Drugs	Targets
NCT03785249	Phase 1/2 Study of MRTX849 in Patients with Cancer Having a KRAS G12C Mutation KRYSTAL-1	I/II	MRTX849 (Adagrasib)	KRAS^G12C^
NCT03948763	A Study of mRNA-5671/V941 as Monotherapy and in Combination with Pembrolizumab (V941-001)	I	mRNA-5671/V941 Pembrolizumab	KRAS^mu^ PD-1
NCT03592888	DC Vaccine in Pancreatic Cancer	I	mDC3/8	KRAS^mu^
NCT04117087	Pooled Mutant KRAS-Targeted Long Peptide Vaccine Combined with Nivolumab and Ipilimumab for Patients with Resected MMR-p Colorectal and Pancreatic Cancer	I	KRAS peptide vaccine Nivolumab Ipilimumab	KRAS^mu^ PD-1 CTLA-4
NCT03745326	Administering Peripheral Blood Lymphocytes Transduced with a Murine T-Cell Receptor Recognizing the G12D Variant of Mutated RAS in HLA-A*11:01 Patients	I/II	Cyclophosphamide Fludarabine Aldesleukin Anti-KRAS G12D mTCR PBL	KRAS^G12D^
NCT03190941	Administering Peripheral Blood Lymphocytes Transduced with a Murine T-Cell Receptor Recognizing the G12V Variant of Mutated RAS in HLA-A*11:01 Patients	I/II	Cyclophosphamide Fludarabine Aldesleukin Anti-KRAS G12V mTCR PBL	KRAS^G12V^
NCT04330664	Adagrasib in Combination with TNO155 in Patients with Cancer (KRYSTAL 2)	I/II	MRTX849 (Adagrasib) TNO155	KRAS^G12C^ SHP2
NCT04185883	Sotorasib Activity in Subjects with Advanced Solid Tumors with KRAS p.G12C Mutation (CodeBreak 101)	I/II	Sotorasib AMG 404 Trametinib RMC-4630 Afatinib Pembrolizumab Panitumumab Carboplatin, pemetrexed, docetaxel, paclitaxel Atezolizumab Everolimus Palbociclib MVASI^®^ (bevacizumab-awwb) TNO155 FOLFIRI FOLFOX BI 1701963	KRAS^G12C^ PD-1 MAP2K1 SHP2 EGFR PD-L1 mTOR CDK4/6 VEGF SOS1

PD-1—programmed cell death protein 1; CTLA-4—cytotoxic T-lymphocyte antigen 4; SHP2—Src homology region 2-containing protein tyrosine phosphatase 2; MAP2K1—mitogen-activated protein kinase kinase; EGFR—epithelial growth factor receptor; PD-L1—programmed death ligand 1; mTOR—mammalian target of rapamycin; CDK4/6—cyclin-dependent kinase 4/6; VEGF—vascular endothelial growth factor; SOS1—son of sevenless.

**Table 2 ijms-24-09313-t002:** Registered trials of alternative targets inhibitor combination therapy on https://clinicaltrials.gov, accessed on 6 March 2023.

ClinicalTrials.Gov Identifier	Title	Phase	Drugs	Targets
NCT04111458	A Study to Test Different Doses of BI 1701963 Alone and Combined with Trametinib in Patients with Different Types of Advanced Cancer (Solid Tumors with KRAS Mutation)	I	BI 1701963 Trametinib	SOS1 MAP2K1
NCT03634982	Dose Escalation of RMC-4630 Monotherapy in Relapsed/Refractory Solid Tumors	I	RMC-4630	SHP2
NCT04916236	Combination Therapy of RMC-4630 and LY3214996 in Metastatic KRAS Mutant Cancers (SHERPA)	I	RMC-4630 LY3214996	SHP2 ERK
NCT04000529	Phase Ib Study of TNO155 in Combination with Spartalizumab or Ribociclib in Selected Malignancies	I	TNO155 Spartalizumab Ribociclib	SHP2 CDK4/6 PD-1
NCT03114319	Dose Finding Study of TNO155 in Adult Patients with Advanced Solid Tumors	I	TNO155 EGF816 (nazartinib)	SHP2 EGFR
NCT04670679	A Dose Escalation/Expansion Study of ERAS-601 in Patients with Advanced or Metastatic Solid Tumors (FLAGSHP-1)	I	ERAS-601 Cetuximab Pembrolizumab	SHP2 EGFR PD-1
NCT04121286	A Study of JAB-3312 in Adult Patients with Advanced Solid Tumors in China	I	JAB-3312	SHP2
NCT04528836	First-in-Human Study of the SHP2 Inhibitor BBP-398 in Patients with Advanced Solid Tumors	I	BBP-398 (Formerly known as IACS-15509)	SHP2
NCT04252339	RLY-1971 in Subjects with Advanced or Metastatic Solid Tumors	I	RLY-1971	SHP2

SOS1—son of sevenless; MAP2K1—mitogen-activated protein kinase kinase; EGFR—epithelial growth factor receptor; SHP2—Src homology region 2-containing protein tyrosine phosphatase 2; ERK—extracellular signal-regulated kinase; CDK4/6—cyclin-dependent kinase 4/6; PD-L1—programmed death ligand 1.

## Data Availability

Not applicable.

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
