# Peer review of "KRAS-Dependency in Pancreatic Ductal Adenocarcinoma: Mechanisms of Escaping in Resistance to KRAS Inhibitors and Perspectives of Therapy"

_ijms, 2023, doi:10.3390/ijms24119313_

Round 1
Reviewer 1 Report
In this review the authors explore KRAS-dependency in pancreatic ductal adenocarcinoma (PDAC) and analyse recent data on KRAS signaling inhibitors focusing on how cancer cells establish compensatory escape mechanisms. They consider that mutational activation of KRAS in the earliest precancerous lesions represents the first genetic event leading to invasive pancreatic cancer. Despite this critical role, its high affinity for nucleotide the lack of viable binding pockets for Small-molecule inhibitors have made direct targeting of the RAS protein extremely difficult over the past four decades. In this context, understanding PDAC tumorigenesis is crucial for both the identification of early diagnostic markers and the development of multiple alternative modes of intervention. They concluded that the endless combination possibilities of inhibitors lead to infinite possibilities of mechanisms of escaping and resistance that they have yet to fully understand and overcome in order to definitely win against PDAC.
I have a suggestion and a question:
1) The authors wrote that "... Pancreatic ductal adenocarcinoma remains a real challenge in oncology. We reviewed all the efforts spent in the recent years. A lot of new possibilities are now actual therapies, but more studies are needed to refine these new strategies...". This review follows the Systematic reviews and Meta-Analyses (PRISMA) statement, published in 2009, which was designed to help systematic reviewers to transparently report: i) why the review was conducted, ii) what the authors did and ii) what did they find?.
2) How many reviews have been published about it?. The period covered by the review is not indicated.
The work is interesting.
Author Response
Response to Reviewer 2 Comments
1) The authors wrote that "... Pancreatic ductal adenocarcinoma remains a real challenge in oncology. We reviewed all the efforts spent in the recent years. A lot of new possibilities are now actual therapies, but more studies are needed to refine these new strategies...". This review follows the Systematic reviews and Meta-Analyses (PRISMA) statement, published in 2009, which was designed to help systematic reviewers to transparently report: i) why the review was conducted, ii) what the authors did and ii) what did they find?.
Response 1: We sought to summarize and make the topic clear. We have focused on KRAS-dependency enlighting his key role in PDAC tumorigenisis. We found that KRASmu influences cells at early stages of tumorigenisis in different ways. We have focused on several possible actors that could be targeted to affect KRAS pathway.
2) How many reviews have been published about it?. The period covered by the review is not indicated.
Response 2: In the past 30 years, around 70 reviews on KRAS role in PDAC have been published. We conducted this review because we felt it was necessary to collect the most up-to-date data covering for the past 5 years and create a new comprehensive information base to rely on for future research.

Reviewer 2 Report
Pancreatic ductal adenocarcinoma (PDAC) is a globaly major oncological problem. RAS mutations are very common molecular incidents for initiation of PDAC development. In submitted manuscript the role of KRAS mutation in PDAC was discussed. In unit 1 (Introduction) the authors described the overview of PDAC molecular changes and further in unit 2 showed more details divided in categories, like KRASmu in relation to metabolism, inflammation or epigenetics. In unit 3 inhibitors of KRAS and clinical trials were introduced. The second part of the review concerns mechanisms of escaping in resistance to KRAS inhibitors, alternative targets in KRAS signaling pathways, and KRAS dependency in PDAC.
Current KRAS targeted therapy is not beneficial in PDAC. Thus, this manuscript is important not only in compilation of available data of PDAC molecular changes and therapy, but also shows various mechanisms why KRAS-based treatment is so difficult.
In my opinion, to make the manuscript clear some minor correction should be prepared.
1. Not all sentences are easy to follow and might be re-written, e.g., lines 62-65 (first part of sentence); line 149 (previous work OR in recen years); lines 346-367 (should be devided in two sentences); lines 461-464.
2. ometimes when the authors refer to gene name (e.g., TP53, CDKN2A) I encourage to use italics.
3. In Figure 1 legend the abbrev. TME should be preceded with full name: tumor microenvironment.
4. NF-kappaB - please use greek symbol of kappa, not k
5. Typos, e.g., line 139 im-munomodulatory; line 418 2x a challenge; line 493 mTOR, not MTOR
6. In the main text number of clinical phases are represented by Roman numbering, but in tables by Arabic numbering - please unified to Roman n.
7. Formatting of Table 1 title according to Journal style.
8. Finally, the authors can think about correction of article title. For instance: KRAS-dependency in Pancreatic Ductal Adenocarcinoma: Mechanisms of Escaping in Resistance to KRAS Inhibitors and Perspectives of Therapy
I suggest to check interpunction. For instance: in lines 92-93 first two commas are not necessary or in line 224 better start a new sentence instead of ";". Sometimes dot is missing in abbreviations, e.g., et al. not et al
In line 119 probably should be gene, not genes.
Author Response
Response to Reviewer 2 Comments
In my opinion, to make the manuscript clear some minor correction should be prepared.
- Not all sentences are easy to follow and might be re-written, e.g., lines 62-65 (first part of sentence); line 149 (previous work OR in recen years); lines 346-367 (should be devided in two sentences); lines 461-464.
Response 1: I re-wrote those sentences following your suggestions.
- Sometimes when the authors refer to gene name (e.g., TP53, CDKN2A) I encourage to use italics.
Response 2: I corrected with italics.
- In Figure 1 legend the abbrev. TME should be preceded with full name: tumor microenvironment.
Response 3: I did it.
- NF-kappaB - please use greek symbol of kappa, not k
Response 4: I replaced k with greek symbol of kappa.
- Typos, e.g., line 139 im-munomodulatory; line 418 2x a challenge; line 493 mTOR, not MTOR
Response 5: I corrected the typos.
- In the main text number of clinical phases are represented by Roman numbering, but in tables by Arabic numbering - please unified to Roman n.
Response 6: I unified to Roman n.
- Formatting of Table 1 title according to Journal style.
Response 7: I did it.
- Finally, the authors can think about correction of article title. For instance: KRAS-dependency in Pancreatic Ductal Adenocarcinoma: Mechanisms of Escaping in Resistance to KRAS Inhibitors and Perspectives of Therapy
Response 8: I really appreciate your suggestion and I have corrected it, thank you.
Comments on the Quality of English Language
I suggest to check interpunction. For instance: in lines 92-93 first two commas are not necessary or in line 224 better start a new sentence instead of ";". Sometimes dot is missing in abbreviations, e.g., et al. not et al
In line 119 probably should be gene, not genes.
Response: I cheched interpuncion and corrected what you reported, thank you.

Round 2
Reviewer 1 Report
In this review the authors explore KRAS-dependency in pancreatic ductal adenocarcinoma (PDAC) and analyze recent data on KRAS signaling inhibitors focusing on how cancer cells establish compensatory escape mechanisms. They consider that mutational activation of KRAS in the earliest precancerous lesions represents the first genetic event leading to invasive pancreatic cancer. Despite this critical role, its high affinity for nucleotide the lack of viable binding pockets for Small-molecule inhibitors have made direct targeting of the RAS protein extremely difficult over the past four decades. In this context, understanding PDAC tumorigenesis is crucial for both the identification of early diagnostic markers and the development of multiple alternative modes of intervention. They concluded that the endless combination possibilities of inhibitors lead to infinite possibilities of mechanisms of escaping and resistance that they have yet to fully understand and overcome in order to definitely win against PDAC.
I had a suggestion and a question to the original manuscript:
1) The authors wrote that "... Pancreatic ductal adenocarcinoma remains a real challenge in oncology. We reviewed all the efforts spent in the recent years. A lot of new possibilities are now actual therapies, but more studies are needed to refine these new strategies...". This review follows the Systematic reviews and Meta-Analyses (PRISMA) statement, published in 2009, which was designed to help systematic reviewers to transparently report: i) why the review was conducted, ii) what the authors did and ii) what did they find?.
The new version of the manuscript includes answers to the previous suggestion and question. I believe that the manuscript has been improved.